# A self-improving triboelectric nanogenerator with improved charge density and increased charge accumulation speed

Li Cheng [1], Qi Xu[2], Youbin Zheng [3], Xiaofeng Jia[2] & Yong Qin [2]

Charge density is one of the most important parameters of triboelectric nanogenerators since it directly determines performance; unfortunately, it is largely restricted by the phenomenon of air breakdown. Here, we design a self-improving triboelectric nanogenerator with improved charge density. A maximum effective charge density of 490 $\mu$C m$^{-2}$ is obtained, which is about two times higher than the highest reported charge density of a triboelectric nanogenerator that operates in an air environment. At the beginning of the working process, the charge accumulation speed is increased 5.8 times in comparison with a triboelectric nanogenerator that is incorporated into the self-improving device. The self-improving triboelectric nanogenerator overcomes the restriction of air breakdown and exhibits an increased effective charge density, which contributes to the improvement of the output performance, and the increase of charge accumulation speed will accelerate the increase of the output power at the start of operation.

[1] Zhongyuan University of Technology, Zhengzhou, Henan 450007, China. [2] Institute of Nanoscience and Nanotechnology, Lanzhou University, Gansu 730000, China. [3] State Key Laboratory of Solid Lubrication, Lanzhou Institute of Chemical Physics, Chinese Academy of Sciences, Lanzhou 730000, China. Correspondence and requests for materials should be addressed to Y.Q. (email: qinyong@lzu.edu.cn)

The development of portable electronics, implantable devices and wireless sensor networks requires long-life and renewable power sources[1]. A triboelectric nanogenerator (TENG)[2–11] holds promise as a viable power source for those small-size devices and networks due to low cost, attractive performance, and easy fabrication[12,13].

In practical applications of a TENG, the generated energy is usually stored in a capacitor or a battery to power functional equipment. Energy storage speed mainly depends on the working frequency and the transferred charge. Since TENGs are driven by vibrations that exist in the environment and the frequencies of such vibrations are usually fixed and cannot be artificially controlled, it is impractical to increase the energy storing speed by increasing the working frequency of TENGs. Hence, the energy storing speed is mainly increased by increasing the charge density. Moreover, the charge density determines the output voltage, current density, and power density of TENGs[14], which is important for application. Several efforts to increase the charge density of TENGs have focused on the choice of materials in the triboelectric series[15], introducing rough surfaces[16,17] and artificially injecting charge into the device[18,19]. Previous attempts have been made to improve the charge density of TENGs, and representative results are listed in Supplementary Table 1. However, when the charge density reaches a very high value, the generated electric field will induce air breakdown; thus, the maximum charge density of TENGs working in air environment has been limited to about 250 μC m$^{-2}$. Accordingly, air breakdown is a major barrier for the improvement of charge density in TENGs. In order to avoid the restriction of air breakdown, a TENG has been placed in high vacuum to increase the charge density to the highest value of 1003 μC m$^{-2}$[12]. Though this new method shows obvious merit in improving the charge density of TENGs, the high vacuum environment is difficult to deploy in practical application of TENGs. Hence, it is significant to explore and develop new technologies that are suitable for fabricating TENGs that operate in an air environment with high charge density.

Besides the limitation of charge density, there is another problem that weakens the performance of TENGs. When a TENG stops working, the triboelectric charge density will slowly attenuate due to water vapor and charged particles that exist in air[20]. So, if a TENG stops working for some time, the output of the TENG at the beginning of its working process will be much lower than that of a TENG that is continuously operating. In some important applications, such as vibration monitoring[21–23] and powering some active sensors[24,25], TENGs need to work for only a few seconds at a time and then remain in standby for hours or more, which requires fast charge accumulation at the beginning of the working process as well as high output within the first few cycles.

Here, we design a self-improving TENG (SI-TENG) to avoid air breakdown and to increase the charge density and charge accumulation speed of TENGs. As a result, we obtain a maximum effective charge density of 490 μC m$^{-2}$, which is the highest charge density of TENGs working in an air environment to the best of our knowledge. The output charge of a SI-TENG is further increased by connecting a couple of part II devices, containing polyethylene terephthalate (PET) films and insulated electrodes, in one device to make a multilayer SI-TENG. Furthermore, the charge accumulation speed at the beginning of the working process of the SI-TENG is improved to 5.8 times of the TENG connected in the SI-TENG, which has significant advantages in intermittent working applications. This simple and highly effective technique to improve TENG performance in common air environments has the potential to improve the effective charge density and charge accumulation speed of other kinds of TENGs, such as TENGs working in sliding[26,27] and spinning[11] modes.

## Results

**Design and structure of the self-improving TENG.** In the working process of TENGs, relative movement of triboelectric charges is the key point to generate electricity in the external circuit. So if we replace the friction layers with new structure which takes larger quantities of charges to make a new kind of device, the performance of TENG will be greatly enhanced. As capacitors can take huge quantity of charges under external voltage, and the plane-parallel capacitor has a similar structure with the friction layers, we design a SI-TENG with inner plane-parallel capacitor structure (PPCS) to greatly enhance the output performance of TENG.

As Fig. 1a, b shows, the SI-TENG contains two parts. Part I is a TENG working in vertical contact-separation mode with polyvinylidene fluoride (PVDF) and polyamide-6 (PA-6) film as the friction layers, respectively. This part is used for generating high voltage under vibration, and filling charge into the PPCS. Part II contains two PET films with two electrodes insulated by PVDF/epoxy resin (EP) films on each film. In this part, electrodes 1 and 2 and the covered PVDF/EP films form the PPCS. The charges generated by part I will be stored in the PPCS to form high charge density in the device. The electrodes 3 and 4 are connected with the external circuit, exporting electrical energy when the device is driven by vibration. The two parts are connected with a rectifier bridge to convert the alternating current (AC) voltage generated by part I to direct current (DC) voltage, and prevent the charge stored in the PPCS from flowing back and neutralizing in the circuit. (Details of the fabrication process of the SI-TENG are shown in the Methods and the reasons for choosing these materials to fabricate the SI-TENG are shown in the Supplementary Note 1.)

**Working mechanism of the self-improving TENG.** The working process of the SI-TENG could be divided into two stages. In the first stage, part I acts as a voltage source to fill charge into the PPCS, as Fig. 1c shows, when the SI-TENG is pressed and released periodically, as an ordinary TENG working in contact-separation mode, part I generates charge on its friction layers and leads to two opposite voltage (or current) peaks in one cycle. The AC current generated by part I is rectified to DC current by the rectifier bridge and filled into the PPCS twice in each cycle. Because the rectifier bridge prevents the charge in the PPCS flowing back, charge will be stored in the PPCS. This process continues until voltage of the charge stored in the PPCS equals to the voltage of part I. In the second stage, when the device is pressed and released under vibration, as the role of the charge on the friction layers in an ordinary TENG, the charges stored in the PPCS generate a changing electric field, which changes the potential difference between electrodes 3 and 4 and pushes electrons flowing between these two electrodes to generate AC voltage (or current) in the circuit.

Based on the working mechanism of the SI-TENG, we theoretically studied the working process of the SI-TENG and obtained the equation of its output charge density with the charge density and voltage of part I (detailed deducing process is given in the Supplementary Note 2):

$$\Delta\sigma \approx \frac{\varepsilon d_2}{\varepsilon_2 d}\sigma_0 \qquad (1)$$

$$\Delta\sigma \approx \frac{\varepsilon\varepsilon_0}{2d}V \qquad (2)$$

In this equation, $\Delta\sigma$ is the output charge density of SI-TENG, $\sigma_0$ is the charge density of part I, $\varepsilon$ and $\varepsilon_2$ are the permittivity of the insulating films in the PPCS and PET film, respectively, in part I,

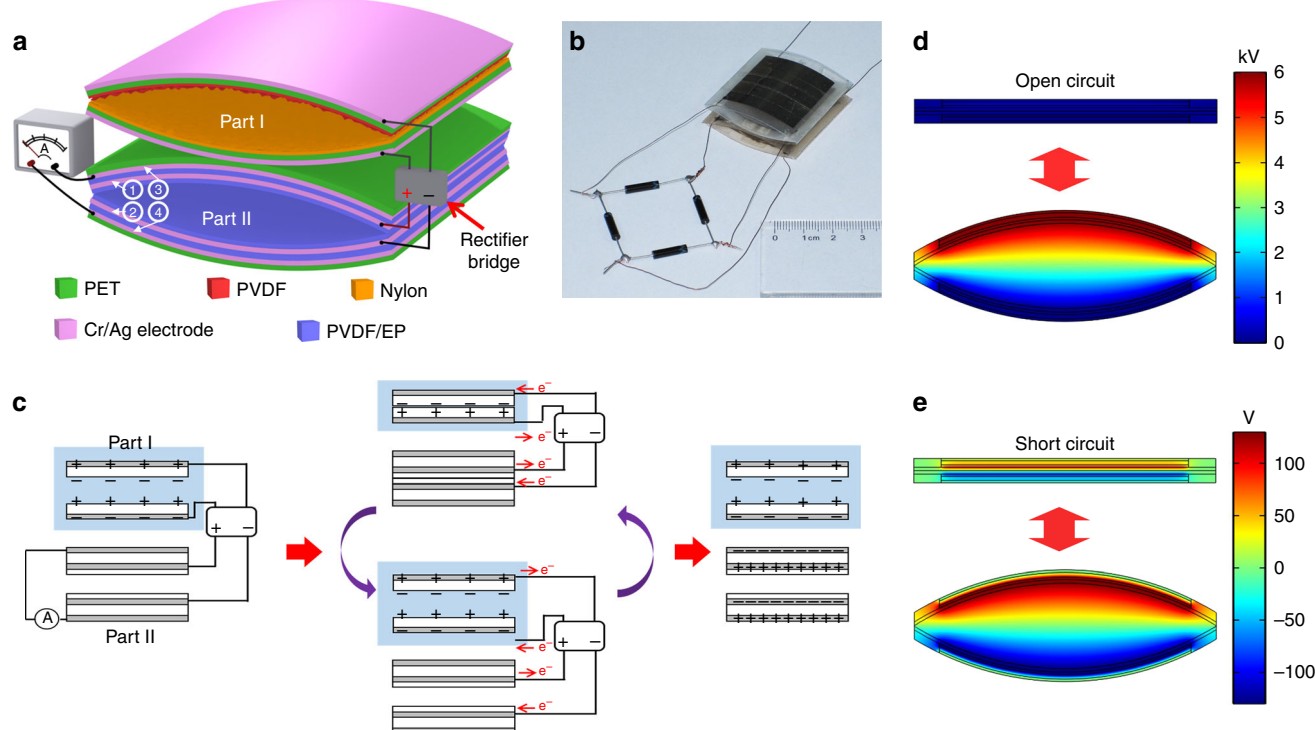

**Fig. 1** Design, structure, and working mechanism of the self-improving triboelectric nanogenerator. **a** Schematic diagram of the self-improving triboelectric nanogenerator (SI-TENG) composed of polyethylene terephthalate (PET), polyvinylidene fluoride (PVDF), and PVDF/epoxy resin (EP). **b** Optical image of a SI-TENG. **c** Charge accumulating process of the SI-TENG and corresponding charge transfer direction in the circuit. The friction layers in part I are replaced by positive and negative charges, and the PET film in part II are not shown in the image. **d**, **e** Simulation results of potential distribution in part II in the open-circuit (**d**) and short circuit (**e**) condition with the gap distance changes between 0 and 340 μm, and the charge density of 300 μC m$^{-2}$ in the plane-parallel capacitor structure (PPCS)

$d$ and $d_2$ are the thickness of the insulating films in the PPCS and PET film, respectively, in part I, $\varepsilon_0$ is the permittivity of vacuum, and $V$ is the output voltage of part I. From these equations, we can see that, if the thickness of the insulating films ($d$) is much smaller than that of the PET films ($d_2$), the charge density will be greatly increased. In addition, a series of parameters such as the output voltage of part I, permittivity and thickness of the insulating films could influence the performance of SI-TENGs. Theoretically study also shows, the insulating layers existing on the surface of PPCS form a distance between the positive and negative charge at the pressed state, and further cause the nonzero voltage and charge density of the measuring electrode at the pressed state, thus the charge density transferred in the circuit is lower than that stored in the PPCS. Since only the charge density transferred in the circuit is useful for powering functional devices in the external circuit, we studied this value in the experiment and defined it as the effective charge density.

To better understand the energy generating mechanism in part II of the SI-TENG, the finite element method calculations of part II under the open-circuit and short-circuit conditions are carried out by using the COMSOL software. In the calculation, the width of the device is set as 1200 μm, the width of the electrodes is set as 900 μm, the thicknesses of the PVDF/EP films are set as 17.2 μm and 14.5 μm (measured in the cross-section of the scanning electron microscopy image of part II, which is shown in Supplementary Fig. 7), and the dielectric coefficients of PVDF/EP films are set as 4.6 (measurement of dielectric coefficients are shown in the Methods), the charge density in PPCS is set as 300 μC m$^{-2}$, and the gap between the PPCS changes from 0 to 340 μm. Under the open-circuit condition (Fig. 1d), the increase of the gap leads to the increase of the voltage between the measuring

electrodes, and the potential drop created by the charges stored in the PPCS increases from 200 to 5999 V. Under the short-circuit condition (Fig. 1e), the increase of the gap makes charges move between the measuring electrodes, and the charge density of the measuring electrode increases from 34 to 289 μC m$^{-2}$, generating 255 μC m$^{-2}$ charge transferred in the circuit.

The SI-TENG was then driven by a linear motor with the amplitude of 16 mm and frequency of 3.33 Hz, and the process of charges filled into the PPCS was studied firstly to verify the working mechanism of SI-TENG. Figure 2a shows the current flowing from the positive pole of the rectifier bridge into the PPCS. We can see the output AC current of part I is converted to DC current by the rectifier bridge, and the decreasing trend of current indicates the voltage between two electrodes of PPCS increasing with the charges filled into PPCS. After enough cycles, the output voltage of the rectifier bridge equilibrates with the PPCS, and the output current of the rectifier bridge decreases to near zero, and the charge density stored in the PPCS reaches the maximum value. Figure 2b, c and Supplementary Movie 1 and 2 show the output voltage of the SI-TENG and the current change with the process of charge filled in the PPCS. The output increases rapidly in the first ten cycles and then increases slowly to a steady value in the following tens of cycles, which is perfectly consistent with the variation of the output current of the rectifier bridge. In the first two cycles, we can clearly see the negative peaks produced by charges filled into the PPCS. Furthermore, as shown in Supplementary Fig. 1, when the connection of the rectifier bridge to the PPCS is reversed, the charges stored in the PPCS reverse its signal, resulting in the reverse of the signal of the SI-TENG. These results further prove that our designed working mechanism of the SI-TENG is correct and effective.

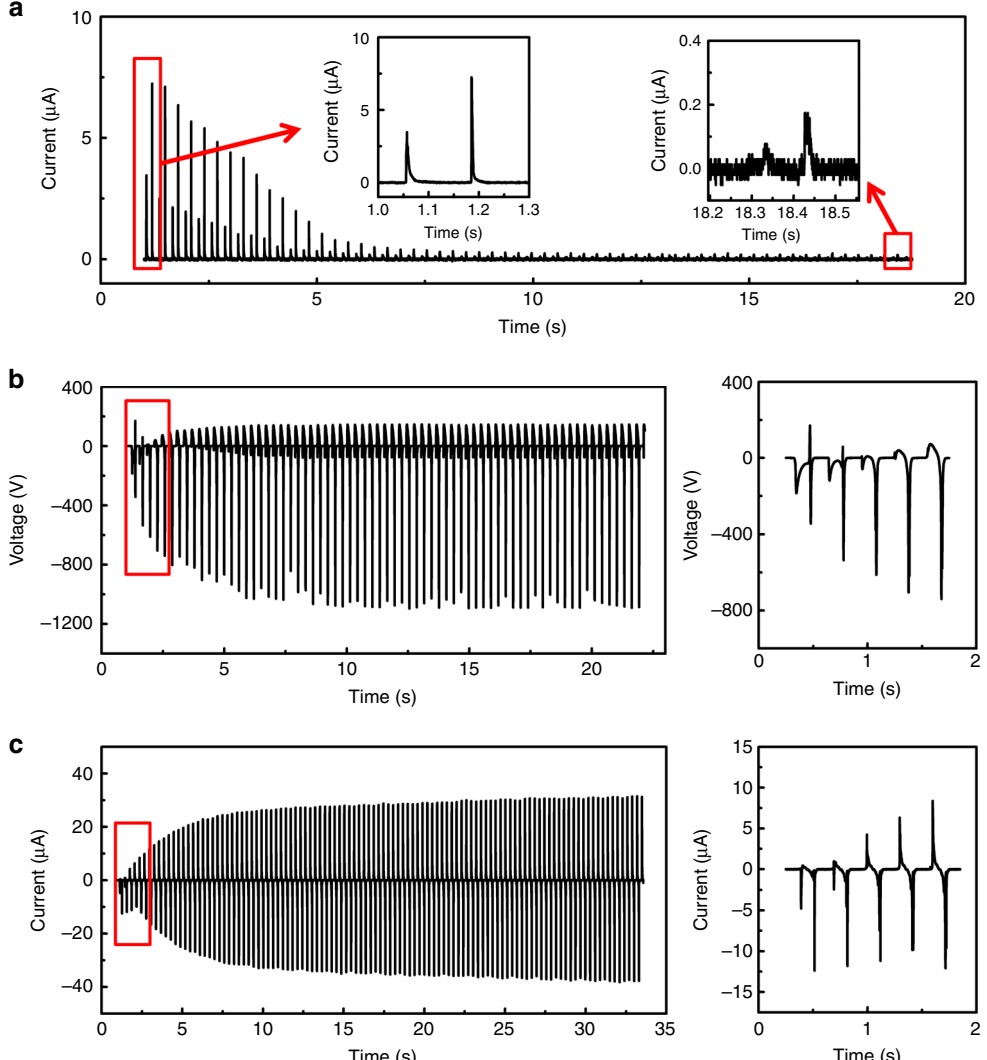

**Fig. 2** Charge accumulation process of the self-improving triboelectric nanogenerator. **a** The current measured at the positive pole of the rectifier bridge shows the charge generated by the voltage source filled into the plane-parallel capacitor structure (PPCS). Output voltage (**b**) and current (**c**) of the self-improving triboelectric nanogenerator (SI-TENG) increase with charge accumulation in the PPCS

**Improvement of effective charge density and output**. In order to quantitatively evaluate the performance of the SI-TENG, we measured the output of part I, part II, and the SI-TENG after the charge density is stable (Fig. 3a–c). Without charge filling by part I, the output current, effective charge density and voltage of part II are 0.1 μA, 0.2 μC m$^{-2}$, and 4 V, respectively, which could be ignored compared with the output of the part I and the SI-TENG. Comparing with the output of part I, the output current and voltage of SI-TENG are respectively increased from 5.6 μA and 460 V to 57.8 μA and 922 V, and the effective charge density is increased from 45 μC m$^{-2}$ to 325 μC m$^{-2}$. These results clearly show the performances of SI-TENG are extensively improved, and the effective charge density is increased to a quite high value, which is higher than the reported charge densities of TENGs working in air condition (about 240 μC m$^{-2}$)[18]. Furthermore, the SI-TENG shows excellent robustness of working continuously for 150,000 cycles without any decrease in performance (shown in Fig. 3d), and even shows a slight increase because the charge density of part I increases slowly at the beginning.

As the charge providing unit in the SI-TENG, part I greatly influences the performance of SI-TENG. Eq. 2 clearly shows increasing the output voltage of part I should be an effective method

to improve the performance of SI-TENG. Through adjusting the output voltage and output current of part I (see Methods), we quantitatively studied how the output current and effective charge density of the SI-TENG change with the output of part I. Supplementary Fig. 2 shows the output current of SI-TENG changes with the voltage of part I ranging from 62 V to 310 V with a step of 62 V. We can clearly see that the output current increases with the voltage of part I, and the statistical result (Fig. 4a) shows both the output current and effective charge density of the SI-TENG have a linear relationship with the voltage of part I. On the other hand, if we keep the voltage of part I unchanged while changing its output current, the output voltage and the effective charge density of the SI-TENG change little (shown in Supplementary Fig. 3 and Fig. 4b), these results fit well with the conclusion drawn from Eq. 2 that the charge density stored in the PPCS is related with the voltage of part I, and increasing the voltage of part I is an effective way to improve the performance of the SI-TENG.

Charge injection has been verified as an efficient method to improve the performance of TENGs, and subsequently we tried to increase the output voltage of part I with a charge injection method reported in our previous work[19] to further increase the output and effective charge density of SI-TENG. As

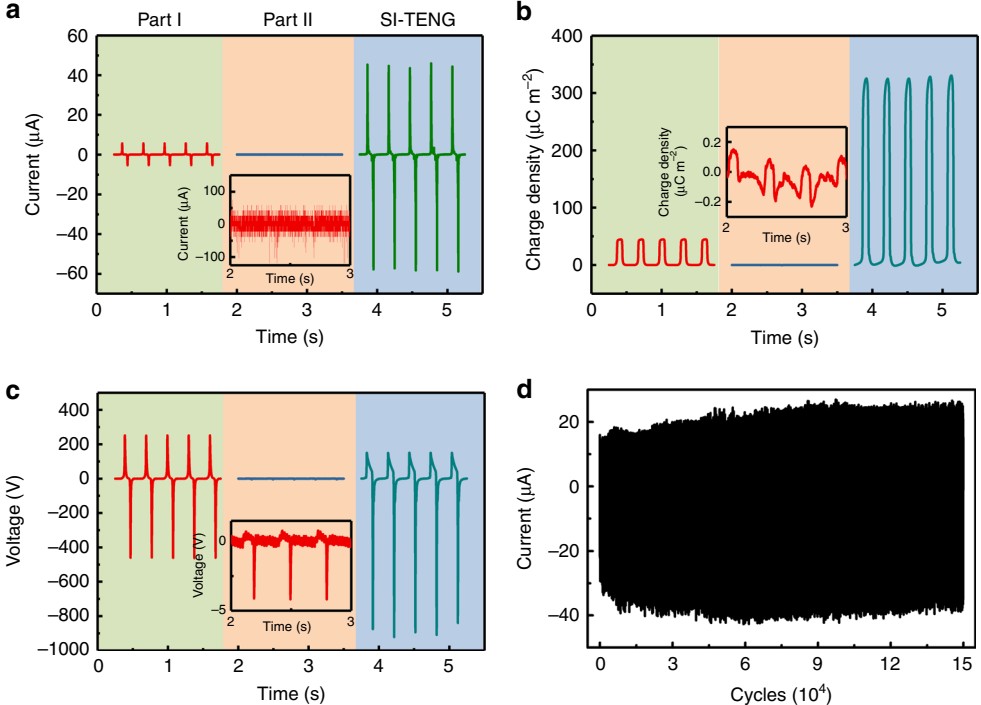

**Fig. 3** Output performance of the self-improving triboelectric nanogenerator. **a–c** Show the output current, the transferred charge density and the output voltage, respectively, of part I (left lines), part II (middle lines), and the self-improving triboelectric nanogenerator (SI-TENG) (right lines). **d** Output current of the SI-TENG working continuously for 150,000 cycles at 3.33 Hz frequency

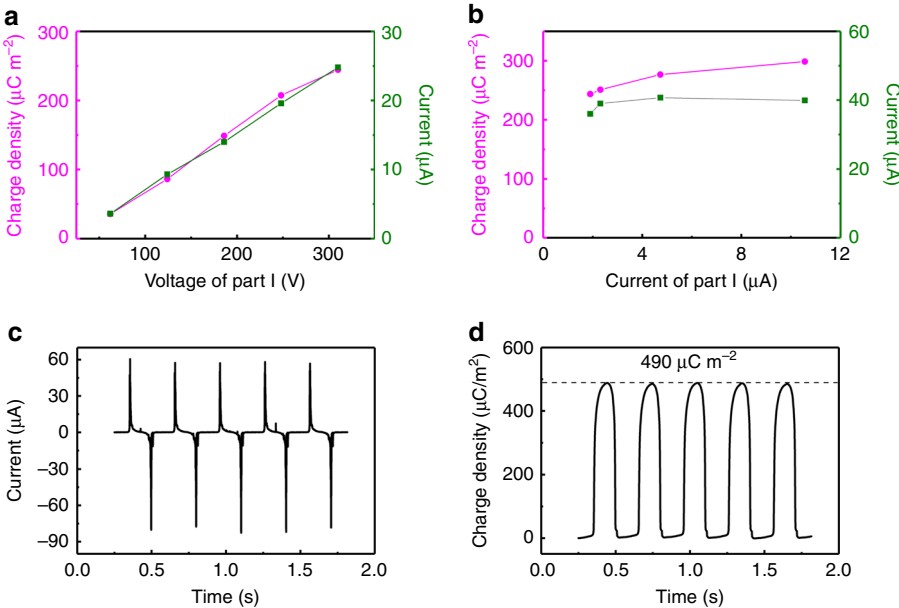

**Fig. 4** Triboelectric nanogenerator performance and part I output. **a**, **b** Show the current (green) and effective charge density (magenta) of self-improving triboelectric nanogenerator (SI-TENG) with the voltage and current of part I. **c**, **d** Show the output current and effective charge density of the SI-TENG, respectively, after further increasing the voltage of part I by the charge injection method

Supplementary Fig. 4 shows, after charge injection, the output voltage of part I is improved to 1140 V (1.65 times of the voltage before charge injection). As a result, as Fig. 4c, d show, after charge injection, the output current of the SI-TENG is improved to 82.6 µA, which is 1.9 times of the output current before charge injection. And the effective charge density is improved to 490 µC m$^{-2}$, which is 1.5 times of the effective charge density before charge injection, this value is about twice of the largest reported

charge densities of TENGs that work in air[18]. The increase of effective charge density will greatly improve the output current, voltage and power of the TENG. Meanwhile, the increase of effective charge density of the TENG directly improves the speed of output energy stored in capacitors or rechargeable batteries, which has great significance in the practical application of TENGs, such as powering personal electronics[28,29], sensors[25,30] and electrochemical systems[19,31].

**Multilayer self-improving TENG**. Since the output voltage of part I keeps constant, output current of part I has little influence on the performance of the SI-TENG. It is possible to fabricate a multilayer SI-TENG to charge a couple of part II devices with only one part I, to obtain higher output in one SI-TENG. So we made four different devices with the same part I and different part II devices (marked as A, B, C, and D), and measured the output current for each of them. As Supplementary Fig. 5 shows, each device generates the current of 40–60 µA and a charge of 250–300 nC for each peak. Afterward, we measured multilayer SI-TENGs charged two, three, and four part II devices with one part I. Supplementary Fig. 5 shows, compared with SI-TENG with one part II, the output of the multilayer SI-TENG is enhanced. Figure 5 shows the statistical result of the output of the multilayer SI-TENG and the sum of corresponding output of the SI-TENG. Since the move of part II devices in the multilayer SI-TENG are not completely synchronized, the output current of multilayer SI-TENG is lower than the sum of corresponding SI-TENGs. The output charge of multilayer SI-TENG is almost equal to the sum of corresponding SI-TENGs. These results indicate that, by connecting several part II devices into one SI-TENG device, output can be further increased.

**Performance enhancement of charge accumulation speed**. As the triboelectric charge existing on the surface of TENGs will slowly neutralize after TENG stops working, the output of TENGs when they start working after a longtime standing is very low. So, at each beginning of TENGs working process, the output power is not high enough to power the attached functional devices. Therefore, the charge accumulation speed of TENGs at the beginning of the working process is very important for application of TENGs, especially for those TENGs working momentarily, such as vibration monitoring[21–23] and powering some active sensors[24,25]. In the SI-TENG, at the beginning of its working process, the PPCS could gather the output charges of part I in several cycles together, and may substantially increase the charge accumulation speed at the beginning of its working process. As shown in Fig. 6a, c, in the first 30 s of the working process of TENG, the output current of the SI-TENG increases from 0 to 32 µA, its increasing speed is about 10 times of part I (increases from 0 to 3.1 µA in 30 s). It can be seen from Fig. 6b, d, the effective charge density of the SI-TENG increases from 0 to 230 µC m$^{-2}$, the charge accumulation speed is about 5.8 times of part I (increases from 0 to 40 µC m$^{-2}$ in 30 s). The increase of charge accumulation speed in SI-TENG is critically important to

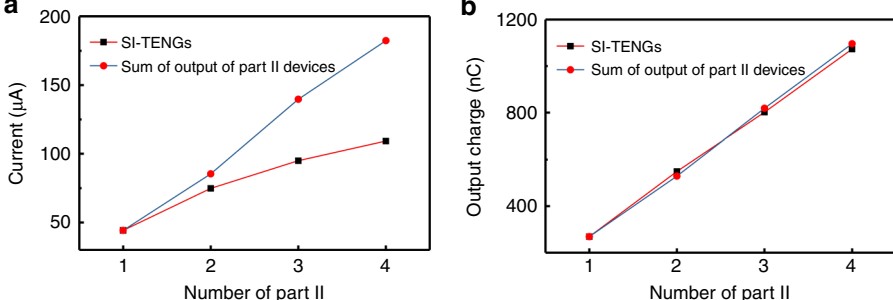

**Fig. 5** Output current and charge of multilayer self-improving triboelectric nanogenerators. Output current (**a**) and charge (**b**) of the multilayer self-improving triboelectric nanogenerators (SI-TENGs) with different numbers of part II devices and the sum of the output of corresponding part II devices working independently

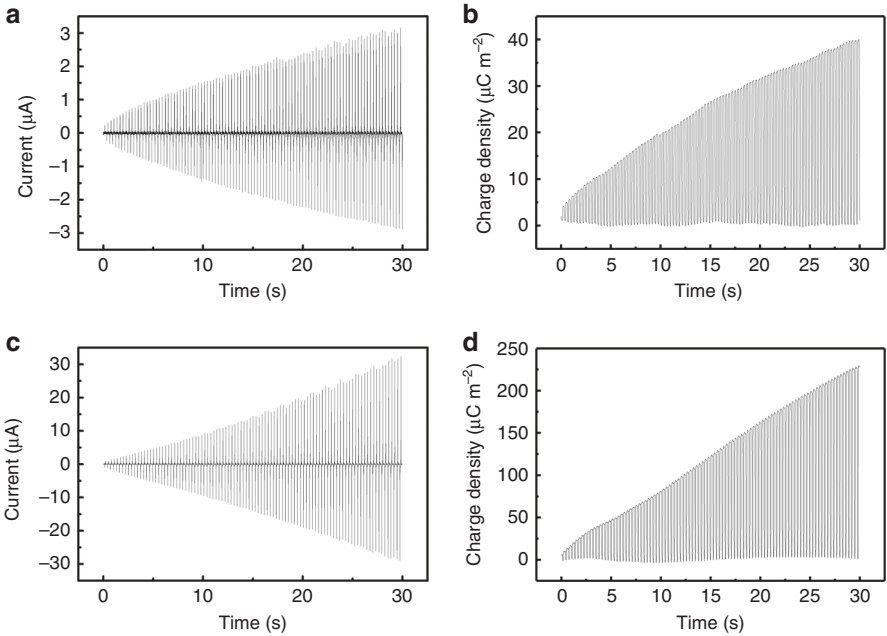

**Fig. 6** Performance enhancement of charge accumulation speed. **a**, **b** Show the output current and effective charge density of part I at the beginning of the charge accumulation process. **c**, **d** show the output current and effective charge density of the self-improving triboelectric nanogenerator (SI-TENG) at the beginning of the charge accumulation process

power the functional devices faster at the beginning of operation, and beneficial to applications especially in the situation where TENGs work momentarily or intermittently.

## Discussion

In summary, we designed and developed a new SI-TENG with improved performance, bypassing the limitation of air breakdown that restricts increased charge density. The output voltage, current and effective charge density are increased to 2.00, 10.32, and 7.22 times that of part I, and a maximum effective charge density of 490 μC m$^{-2}$ is obtained, which is about twice that of the highest reported charge density for a TENG working in air, to the best of our knowledge. By fabricating multilayer SI-TENGs with multiple part II devices in each TENG device, the output charge can be increased to the sum of the output of each part II device. Meanwhile, the charge accumulation speed is increased to 5.8 times that of an ordinary TENG. The increase of charge density and charge accumulation speed is attractive for applications of TENGs. Easy fabrication and wide adaptability to working environment make this technique for the development of SI-TENGs suitable for expansion to other kinds of TENGs. Furthermore, our theoretical analysis provides three ways to further increase the effective charge density of SI-TENGs: increasing the output voltage of part I, replacing the insulating films with other high permittivity materials, and reducing the thickness of the insulating films. In this work, we have increased the effective charge density and output current through a SI-TENG design. This technique and finding will significantly facilitate the future studies on SI-TENGs and their applications.

## Methods

**Preparation of PVDF and polyamide-6 solutions**. The PVDF solution is prepared by adding 3.75 g PVDF, 8.5 g N,N-dimethylacetamide (DMAC) and 12.75 g acetone into a 50 mL triangular flask, stirring at 60 °C for 30 minutes and cooling to room temperature to make PVDF completely dissolve. The PA-6 solution is prepared by adding 2 g PA-6 and 18 g formic acid into a 50 mL triangular flask, and stirring until PA-6 completely dissolves. All reagents are analytically pure and used without any further purification.

**Fabrication of the self-improving TENG**. As shown in the schematic diagram in Supplementary Fig. 6a, two pieces of cleaning PET film with the thickness of 150 μm are cut into 4 cm × 4 cm, and Cr/Ag electrodes with the size of 3 cm × 3 cm are sputtered on the middle of one side of the PET films. On the other side of PET films, PVDF solution or PA-6 solution is spin-coated respectively, on one piece of PET films with the speed of 5000 rpm for 30 s, the films are then drying in room temperature. After the connection of copper conductors with the electrodes, the PET films are fixed together with PVDF film contacting with the PA-6 film. The device is then heated to 80 °C under bending state and cooled to room temperature to make the device arch shaped. This device forms part I of the SI-TENG.

As shown in the schematic diagram in Supplementary Fig. 6b, two pieces of clean PET films with the thickness of 150 μm are cut into 4 cm × 4 cm, and Cr/Ag electrodes with the size of 3 cm × 3 cm are sputtered on the middle of one side of the PET films, the electrodes are then connected to the back of the PET films with carbon paste. After this, PVDF solution is spin-coated on the electrodes at 3000 rpm for 30 s, and dried at room temperature for 30 min. Then about 0.5 mL EP solution is smeared evenly on the PVDF film, and the bubbles in the PVDF film and the EP solution are removed by placing them in vacuum for 5 min. Then the EP solution is spin-coated at 3000 rpm for 60 s and heated at 60 °C for 2 h to fully cure the EP solution. Then, another Cr/Ag electrodes with the size of 3 cm × 3 cm are sputtered on the PVDF/EP films, after the connection of the electrodes to the back of the PET films with carbon paste, another PVDF/EP films are fabricated on the electrodes as the procedure shown above, except that the spin-coating speed of the EP solution is changed to 6000 rpm to make the film thinner. After this, copper conductors are connecting on the carbon pastes on the back of the PET films, which connected with the electrodes. At last, the films are fixed together with the PVDF/EP films at the inside, and the device is heated to 80 °C under bending state and cooled to room temperature to make the device arch shaped. This device forms part II of the SI-TENG.

Two parts of the SI-TENG are fixed together and connected with the rectifier bridge as the schematic diagram shown in Fig. 1a. The thicknesses of the PVDF/EP films are 17.2 and 14.5 μm (shown in Supplementary Fig. 7).

**Measurement of dielectric coefficient of PVDF/EP films**. The dielectric coefficients of two PVDF/EP films are measured. The dielectric coefficient of the film sandwiched by two electrodes (electrodes 1 and 3 or electrodes 2 and 4 shown in Fig. 1a) is measured by adding triangular wave voltage with 10 V voltage and 0.1 Hz frequency between the two electrodes, measuring the current in the circuit and calculating the capacitance on the electrodes. As the thickness of PVDF/EP films are negligible compared with the length and width of the electrodes, the capacitor can be regarded as a dielectric coefficient, and the dielectric coefficient of the film can be calculated from the equation:

$$\varepsilon = \frac{dC}{\varepsilon_0 S}$$

The dielectric coefficient of the PVDF/EP film covering the surface of electrodes 1 or 2 is measured with the same method after sputtering another electrode on the surface to sandwich the film by two electrodes. As a result, dielectric coefficients of both the PVDF/EP films are 4.6.

**Control of the output voltage and current of part I**. To control the output voltage of part I, a series of 62 V zener diodes are connected with the positive pole together in pairs. The output voltages of the TENGs are set at 62, 124, 186, 248, and 310 V by connecting 1 to 5 pairs of zener diodes in series and then connecting in parallel with the TENG. The output currents of part I are controlled by cutting the device to a smaller size.

## Data availability:

The data that support the findings of this study are available from the corresponding author upon reasonable request.

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

## Acknowledgements

We sincerely acknowledge the support from NSFC (NO. 51702377, 51472111).

## Author contributions

L.C. and Y.Q. designed the SI-TENG, L.C. and X.J. fabricated and measured the device, Q.X. conducted the simulation via COMSOL, L.C., Y.Z., and Y.Q. analyzed the experimental data, plotted the figures and prepared the manuscript.

## Additional information

**Competing interests:** The authors declare no competing interests.

