## [Peer Review File · Nature Communications]

Reviewers' comments:

Reviewer #1 (Remarks to the Author):

The manuscript provided an innovative approach to improve the charge density and the charge accumulation speed of a triboelectric nanogenerator (TENG). The new method of integrating a capacitor and increasing the voltage of the TENG in part I in the manuscript can greatly improve the energy density and power density of the energy harvesting device. This is a milestone work in the field of the nanogenerator, and I strongly support the publication of the manuscript. I have only one suggestion for the author to consider. Figure 3d indicates that the current increased with times in the robustness test. The author may consider providing an explanation for the change of current over time.

Reviewer #2 (Remarks to the Author):

This article presents an unique laminating structure of triboelectric nanogenerator with reasonable data regarding a higher surface charge density and fast accumulation speed. Although the manuscript is well written and data presented is reasonable, it needs a major revision to be published as listed followings

Per author's conclusion, one significant achievement is a higher charge density compared to the previous report. However, direct comparison of charge density values with Ref 18 may not be accurate because of their inconsistency of the position of the charge storage layer. Author designed the charge storage layer in between dielectric materials which surely overcome air discharge limits. And calculation of charge density using a very simple equation such as parallel-plate one should be modified. The It should be calculated precisely because TENG presented here used arc-shaped structure with many dielectric materials such as air, PVDF and PVDF/EP. Besides the precise calculation, a proper reference regarding the charge density of the previous reports should be addressed.

Mostly, high output triboelectric device can maintain the peak of open-circuit voltage signal. In the measured voltage data presented here, however it appeared that all voltage peak values dissipated instantly. The reason of that should be presented. And such type of peak voltage is a function of mechanical input of frequency. The manuscript did not contain any detail information of input energy and frequency of mechanical driver.

Accumulation speed was only compared with the author's own device and its value was varied

from Abstract (4.8) to Conclusion (5.8). It must be revised and provided with others' data for the general accumulation time.

There is no proper explanation of the choice of materials. If there are scientific reason for triboelectric device, the author should address it. Otherwise PVDF like a ferroelectric material with somewhat high-cost may not be proper for the cost-effective triboelectric generators.

Reviewer #3 (Remarks to the Author):

Triboelectric nanogenerators (TENGs) have attracted a great deal of attention due to its potential as a power supply for internet of things and small sensors. Improving the output of TENG has huge impact for its practical applications. This paper reports a new idea of how to “pump up” the effective charge density for TENG so that the output can be maximized. The work can be important for practical application as well as fundamental research. Here are my specific questions:

1. The authors have demonstrated an enhancement of a breakdown surface charge density from $250 \mu\text{C}/\text{m}^2$ to $490 \mu\text{C}/\text{m}^2$, which likely to improve the output power. Since the structure is a double TENG integration, the first one is used to pump up the second one. The questions is, is the output of the new double layered TENG more than the sum of the output of the two if both working independently? In other workd, is it $1 + 1 > 2$? Please use data to show the results.
2. Can the authors use one TENG as the pump for a stack of other TENGs in order to boost up their effective surface charge density? Can the author build a 3 layer structure and use one of them to charge up the other two. I think that such a demo can be very important.
3. I like to see photographs of the device so that a good sense is received about its true picture.
4. As for the writing, the authors are required to correct some grammars and make the figure more attractive.
5. I also like to see more video demons of the work so that the readers can appreciate the creativity of the work.
6. The abbreviation of STEN can be SI-TENG

Responses to Reviewers:

Response to reviewer #1

The manuscript provided an innovative approach to improve the charge density and the charge accumulation speed of a triboelectric nanogenerator (TENG). The new method of integrating a capacitor and increasing the voltage of the TENG in part I in the manuscript can greatly improve the energy density and

power density of the energy harvesting device. This is a milestone work in the field of the nanogenerator, and I strongly support the publication of the manuscript. I have only one suggestion for the author to consider. Figure 3d indicates that the current increased with times in the robustness test. The author may consider providing an explanation for the change of current over time.

Response: We thank the referee very much for the positive evaluation and nice suggestion. The change of current was mainly caused by the changes of part I's output. Because part I generates charge on its friction layers via the triboelectrification process. At the beginning of SI-TENG's working process, the charge density will increase slightly, thus, the output voltage of part I and output of SI-TENG will increase at the meantime. We have added a brief explanation in the manuscript.

Response to reviewer #2:

This article presents an unique laminating structure of triboelectric nanogenerator with reasonable data regarding a higher surface charge density and fast accumulation speed. Although the manuscript is well written and data presented is reasonable, it needs a major revision to be published as listed followings.

Response: We thank referee very much for the positive and valuable comments. we have revised our manuscript carefully according to these comments.

Per author's conclusion, one significant achievement is a higher charge density compared to the previous report. However, direct comparison of charge density values with Ref 18 may not be accurate because of their inconsistency of the position of the charge storage layer.

Response: Thank you very much for your comment. In part II of our device, charge was filled into the PPCS by part I, and because there are insulating layers in the PPCS,

the charge transferred in the circuit will be less than the charge stored in the PPCS (this could be seen clearly by the calculation result). Thus, as only the charge density transferred in the circuit is useful for powering functional devices in the external circuit, we studied this value in the experiment and defined it as the effective charge density.

In previously reported TENGs, as there was no insulating layers, the charge transferred in the circuit is almost equal to the charge stored on the friction layers, and researchers calculated the charge density from the integration of current.

Consequently, from the point of the charge density useful for powering functional devices, the effective charge density studied in our work is comparable to the charge density studied in Ref. 18 and other reported TENGs.

Author designed the charge storage layer in between dielectric materials which surly overcome air discharge limits. And calculation of charge density using a very simple equation such as parallel-plate one should be modified. The It should be calculated precisely because TENG presented here used arc-shaped structure with many dielectric materials such as air, PVDF and PVDF/EP.

Response: Thanks very much for your good suggestion. We have recalculated the potential distribution, voltage changes under open-circuit condition and charge distribution under short-circuit condition of part II with a new model closer to the actual device. In our new model, the width of the device is set as 1200 μm , the width of the electrodes is set as 900 μm , the thickness of the PVDF/EP films are set as 17.2 μm and 14.5 μm (measured in the cross-sectional SEM image of part II, which is shown in Supplementary Fig. 7), and the dielectric coefficients of PVDF/EP films are set as 4.6 (this value was measured experimentally, and measurement of dielectric coefficients are shown in the methods).

We have revised the calculation result in the manuscript, and added the measurement process of dielectric coefficients in the methods.

Besides the precise calculation, a proper reference regarding the charge density

of the previous reports should be addressed.

Response: Thanks for your suggestion. We have listed the representative reported results of high charge density TENGs in Supplementary Table 1.

Mostly, high output triboelectric device can maintain the peak of open-circuit voltage signal. In the measured voltage data presented here, however it appeared that all voltage peak values dissipated instantly. The reason of that should be presented. And such type of peak voltage is a function of mechanical input of frequency. The manuscript did not contain any detail information of input energy and frequency of mechanical driver.

Response: Thanks for your questions. Till now, there are two test mode to measure the voltage of TENGs. One mode is to use an electrometer, as its input resistance is very high, the charge will not transfer in the circuit, the voltage will be presented as steps in the result [e.g. *Adv. Mater.* **26**, 6720-6728 (2014)]. The other mode is to use a voltmeter, its input resistance is not high enough to prevent the charge transferring in the circuit, thus, charge will transfer between the electrodes, and the voltage peak dissipated when TENGs are kept at the pressed or released states [e.g. *Nano Lett.* **12**, 3109-3114 (2012)]. The second method is widely used in the past published papers. So to have a good comparison, in this work, the voltage are measured by the second mode, thus, the voltage peak dissipated instantly when the SI-TENG is kept at the pressed and released states.

In the experiment, we use a linear motor to drive the SI-TENG with the amplitude of 16 mm and frequency of 3.33 Hz. We have added this in the revised manuscript.

Accumulation speed was only compared with the author's own device and its value was varied from Abstract (4.8) to Conclusion (5.8). It must be revised and provided with others' data for the general accumulation time.

Response: Thanks for your nice comment. In the manuscript, we used the word "increase by" in the abstract and the word "increase to" in the introduction and discussion, which leads to the difference of the values "4.8" and "5.8". To describe it

more clearly, we have unified the values in the manuscript.

We consulted a large deal of papers about TENGs, and found no reported data about TENGs' charge accumulation speed when TENGs start working, thus, the charge accumulation speed in our work couldn't be compared with a general value. So, in our work, the charge accumulation speed was compared with the value of part I to demonstrate the improvement on charge accumulate speed.

There is no proper explanation of the choice of materials. If there are scientific reason for triboelectric device, the author should address it. Otherwise PVDF like a ferroelectric material with somewhat high-cost may not be proper for the cost-effective triboelectric generators.

Response: Thanks for your good suggestion. Materials used in the SI-TENG are chosen for the following reason. PVDF and PA-6 are chosen as friction layers of part I, because PVDF and PA-6 are respectively highly negative and positive in the triboelectric series, and they are easy to generate negative and positive charge in the friction process. PVDF/EP films are chosen as the insulating films, since dielectric coefficient of PVDF are relatively high in polymers, thus, the capacitance between electrodes 1 and 2 is higher, and the charge density filled into the PPCS could be higher. Because the spin-coated PVDF film is not compact, EP was chosen to make the film compact to avoid the insulating film's breakdown under high voltage. And we have added this in the supplementary information.

The price of 10 kg PVDF powder sold on Alfa Aesar are 12674 CNY (1998 USD), and in our fabrication process, 1 g PVDF could be used for fabricating 20-30 SI-TENGs. Thus, the price of PVDF used in one SI-TENG is about 0.008 USD. It is not too expensive for fabricating SI-TENG.

Response to reviewer #3:

Triboelectric nanogenerators (TENGs) have attracted a great deal of attention

due to its potential as a power supply for internet of things and small sensors. Improving the output of TENG has huge impact for its practical applications. This paper reports a new idea of how to “pump up” the effective charge density for TENG so that the output can be maximized. The work can be important for practical application as well as fundamental research. Here are my specific questions:

Response: We appreciate the referee for the high evaluation of our work and the important comments. We have fabricated and tested some new devices and revised our manuscript carefully according to these comments.

1. The authors have demonstrated an enhancement of a breakdown surface charge density from $250 \mu\text{C}/\text{m}^2$ to $490 \mu\text{C}/\text{m}^2$, which likely to improve the output power. Since the structure is a double TENG integration, the first one is used to pump up the second one. The question is, is the output of the new double layered TENG more than the sum of the output of the two if both working independently? In other words, is it $1 + 1 > 2$? Please use data to show the results.

Response: Thanks for your questions. We respectively measured the output of a new SI-TENG and its corresponding part I and part II. As a result, the output current, effective charge density and voltage of part I are $5.6 \mu\text{A}$, $45 \mu\text{C m}^{-2}$ and 460 V . Without the charge filled by part I, the output current, effective charge density and voltage of part II are $0.1 \mu\text{A}$, $0.2 \mu\text{C m}^{-2}$ and 4 V , which could be ignored comparing with the output of part I and the SI-TENG. The output current, effective charge density and voltage of part I are $57.8 \mu\text{A}$, $325 \mu\text{C m}^{-2}$ and 922 V . This result clearly shows the output of the SI-TENG are much higher than the sum of the output of part I and part II. We have added these data into the manuscript to replace the result shown in Fig. 3a-c and the output of pure part II are discussed in the manuscript.

2. Can the authors use one TENG as the pump for a stack of other TENGs in order to boost up their effective surface charge density? Can the author build a 3 layer structure and use one of them to charge up the other two. I think that such

a demo can be very important.

Response: We thanks the referee for this important comment. This technique is very useful to further increase the output of SI-TENG. We fabricated a new SI-TENG with one part I to charge up 2 or more part II (defined as multilayer SI-TENG in the manuscript). As a result, the output current of multilayer SI-TENGs are increased clearly comparing with the SI-TENG composed of a single part II, However, since the moving of part IIs in the multilayer SI-TENG are not completely synchronized, the output current's peak of multilayer SI-TENG are lower than the sum of corresponding SI-TENGs' peaks with single part II. But the output charge of multilayer SI-TENG are almost equal to the sum of corresponding SI-TENGs with single part II. We have added these result in the manuscript in Fig. 5 and Supplementary Fig. 5, with a description of the multilayer SI-TENG.

3. I like to see photographs of the device so that a good sense is received about its true picture.

Response: Thanks for your comment. We have added the photograph of the SI-TENG in Fig. 1b.

4. As for the writing, the authors are required to correct some grammars and make the figure more attractive.

Response: Thanks for your suggestion. We have carefully corrected the grammars in the manuscript and changed the figures to make them more understandable and attractive.

5. I also like to see more video demons of the work so that the readers can appreciate the creativity of the work.

Response: We thanks very much for your suggestion. We have added two videos in the supplementary information as Supplementary Movie 1 and Supplementary Movie 2 to show how the output current and voltage changes with charge filled into part II.

6. The abbreviation of STEN can be SI-TENG

Response: We thanks for your good suggestion. We have revised all the words “STNG” in the manuscript into “SI-TENG”.

Reviewers' Comments:

Reviewer #1 (Remarks to the Author):

The author revised the original manuscript as suggested by reviewers. However, still, the major scientific issue raised by reviewers and significance of this work may not satisfy the reader of nature communication. The following is an issue the author should address before the publication.

Combining two parts of tirboelectric generators could be effective design for fast charging TENG, but it may not give any new scientific intuition. Author wrote that the output power from the hybrid device surpass largely the sum of powers from each device. However, its detail mechanism and plausible equation should be addressed.

Reviewer #2 (Remarks to the Author):

The authors have fully addressed my questions, and I have no more comments. I think that the paper is in a good shape for publication.

Responses to Reviewers:

Response to reviewer #1

The author revised the original manuscript as suggested by reviewers.

However, still, the major scientific issue raised by reviewers and significance of this work may not satisfy the reader of nature communication. The following is an issue the author should address before the publication. Combining two parts of triboelectric generators could be effective design for fast charging TENG, but it may not give any new scientific intuition. Author wrote that the output power from the hybrid device surpass largely the sum of powers from each device. However, its detail mechanism and plausible equation should be addressed.

Response: We thank referee very much for the comment. We have deduced a theoretical equation and added it in our manuscript to describe the mechanism more deeply. And in the Supplementary Information, we theoretically studied the SI-TENG's working process and gave the detailed deducing process of the this approximate equation describing the relationship of the SI-TENG's effective charge density with part I's charge density or voltage.

Response to reviewer #2

The authors have fully addressed my questions, and I have no more comments. I think that the paper is in a good shape for publication.

Response: We appreciate referee very much for the positive evaluation of manuscript.

Reviewer's Comments

Reviewer #2 (Remarks to Author):

The author thoroughly resolved the issue regarding the working mechanism of the fabricated high performance device. Now, the revised paper is in a good shape for publication.